# Waste Management and the Perspective of a Green Hospital—A Systematic Narrative Review

**DOI:** 10.3390/ijerph192315812

**Published:** 2022-11-28

**Authors:** Sabrina Lattanzio, Pasquale Stefanizzi, Marilena D’ambrosio, Eustachio Cuscianna, Giacomo Riformato, Giovanni Migliore, Silvio Tafuri, Francesco Paolo Bianchi

**Affiliations:** 1Dipartimento di Medicina di Precisione e Rigenerativa e Area Jonica, University of Bari Aldo Moro, Piazza Giulio Cesare 11, 70124 Bari, Italy; 2Interdisciplinary Department of Medicine, University of Bari Aldo Moro, Piazza Giulio Cesare 11, 70124 Bari, Italy; 3Bari Policlinico University Hospital, 70124 Bari, Italy

**Keywords:** healthcare workers, green hospital, operating room, climate change, COVID-19

## Abstract

The concept of a “green hospital” is used in reference to a hospital that includes the environment as part of its quality services and one that pays attention to the sustainable design of buildings. Waste disposal represents a potential risk for the environment; therefore, waste collection from healthcare centers is a key environmental issue. Our study aims to systematically review the experiences acquired in worldwide nosocomial settings related to the management of healthcare waste. Nineteen studies, selected between January 2020 and April 2022 on Scopus, MEDLINE/PubMed and Web of Science databases were included in our systematic narrative review. Operating room and hemodialysis activities seem to be the procedures most associated with waste production. To deal with waste production, the 5Rs rule (reduce, reuse, recycle, rethink and research) was a common suggested strategy to derive the maximum practical benefit while generating the minimum amount of waste. In this context, the COVID-19 pandemic slowed down the greening process of nosocomial environments. Waste management requires a multifactorial approach to deal with medical waste management, even considering the climate change that the world is experiencing. Education of health personnel and managers, regulation by governmental institutions, creation of an “environmental greening team”, and awareness of stakeholders and policymakers are some of the measures needed for the greening of healthcare facilities.

## 1. Introduction

A “green hospital” is referred to a healthcare facility that comprises environment as part of its quality services and cares about the sustainable design of edifices. It requires some features such as strategic location, proficient water usage, energy and decent air pollution and the use of worthy materials. A “green hospital” preserves indoor environmental quality, delivers good food, provides green education, focuses on green products, non-toxic environments, green cleaning, waste reduction, and offers a healing garden [1].

Green healthcare which entails the implementation of environmentally friendly practices into health care delivery, denotes an additional assessment for healthcare workers and institutions. It provides the possibility of preserving the environment, which is a progressively fascinating challenge. It permits healthcare organizations to determine leadership in their societies. It can be a manifesto for teaching students and the general population, and it can also be an appropriate policy to save capital [2].

One of the principal obstacles is the absence of organization in hospitals in terms of handling waste disposal. Waste disposal characterizes a possible risk for the ecosystem; therefore, waste collection at healthcare facilities turns out to be a critical point [3]. Wastes produced during health care activities have a greater possible risk of infection and injury than any other type. Scientific literature indicates that 75–90% of the waste produced in the healthcare nosocomial is deprived of risk if compared with the wastes generated by houses. These wastes are mostly produced by the organizational and managerial functions of these centers and only 10–25% of the total amount of waste is considered dangerous [3]. Many ecological concerns in health areas are directly linked to waste production and dumping methods. World population growth, increased lifetime and global crisis involve more waste production, which requires better management [4].

As reported by the World Health Organization (WHO), the most common issues related to healthcare waste are an unawareness about the health dangers associated with it, inadequate training in proper waste management, the lack of waste administration and disposal systems, insufficient economic and human resources, and poor attention given to the matter [5]. Key elements to improve healthcare waste management are:the promotion of practices that reduce the volume of wastes produced and ensure proper waste segregation;the development of strategies and systems to incrementally improve waste segregation, destruction and disposal practices with the ultimate goal of reaching national and international standards;the adoption of safe and green treatment of dangerous health care wastes (e.g., by autoclaving, microwaving, chemical treatment, etc.) over medical waste incineration;the creation of a comprehensive system, addressing duties, resource distribution, handling and disposal;the raising of awareness for risks related to healthcare waste, and of safe practices;the selection of safe and environmentally friendly organizational options, to protect people from the risks related to their work in terms of collecting, handling, storing, transporting, treating or disposing of waste [5].

In this context, our study aims to systematically review the experiences in nosocomial settings related to the approach and management of healthcare waste, worldwide. We aim to define the procedures most associated with waste production, focusing on the strategies to manage this issue. Moreover, we focus on the impact of COVID-19 on this subject and on the future challenges related to climate change. We analyze the strategies of hospitals and policymakers to deal with healthcare waste and define the strategies suggested by scientific literature to deal with their management. Finally, we focus on possible research gaps on this topic and look for answers in the most recent scientific literature. 

## 2. Materials and Methods

Scopus, MEDLINE/PubMed, and ISI Web of Knowledge databases were systematically searched. Research articles, brief reports, commentaries, and letters published between 1 January 2020 and 30 April 2022, were included in our search. The following key words were used as part of our search strategy: (green hospital) AND (waste). Full-text English studies were included. Abstracts without full text, systematic reviews, meta-analyses, and all studies focusing on issues unrelated to the purpose of this review (air pollution, radiation, intra-hospital mobility, etc.) were excluded. When necessary, study authors were contacted in order to collect additional information. References of all articles were reviewed for further studies. The list of papers was independently screened by title and/or abstract by two reviewers who applied the predefined inclusion/exclusion criteria. Discrepancies were recorded and resolved by consensus.

## 3. Results

### 3.1. Identification of Relevant Studies

The flow chart, constructed following the PRISMA guidelines [6] (Figure 1), shows the process of article selection. According to the aforementioned inclusion criteria, 12 articles were identified in Scopus, 15 in ISI Web of Knowledge and 9 in MEDLINE/PubMed. After excluding duplicate articles in the two databases, there were 25 eligible studies. Among them, six were excluded because they did not fulfill the inclusion criteria. Thus, in total, 19 studies were eligible [7,8,9,10,11,12,13,14,15,16,17,18,19,20,21,22,23,24,25] (Table 1). The remaining 209 studies did not meet the inclusion criteria.

### 3.2. Systematic Review

Many studies were focused on waste management in operating rooms; indeed, operating rooms and procedural suites produce a huge quantity of garbage, accounting for 30–70% of all health care waste. Shum PL et al. [24] performed a waste audit of 17 neurointerventional procedures at an Australian hospital over three months. Waste was classified into five branches: general waste, clinical waste, recyclable plastic, recyclable paper, and sharps. The processes produced 135.3 kg of garbage: 85.5 kg (63.2%) clinical waste, 28.0 kg general waste, 14.7 kg recyclable paper, 3.5 kg recyclable plastic, 2.2 kg recyclable soft plastic, and 1.4 kg of sharps; a mean of 8 kg of garbage was produced per case. In particular, endoscopy services seem to be the second garbage maker in a health facility [15]; to deal with the endoscopy-related waste, authors suggested focusing on supplies (multiple-use devices, recycling single-use ones), minimizing waste in wrapping and endoscopes (multiple-use, recycling) [15].

Vacharathit V et al. [8] reported the experience of the Cleveland Clinic that applied a physician-driven protocol to involve surgical staff and trainees to reduce waste in the operating rooms. Each year, involved surgery residents present a self-driven suggestion to green the operating theaters. The principal suggestion was focused on training concerning appropriate garbage segregation, diversion of pre-incision plastics from garbage to be recycled, and local community organization for supplementary recycling sorting. These interventions resulted in around 1 million pounds of plastics diverted from landfills and regulated medical waste was decreased by 26 tons per month for reference.

Gill AS et al. [19] focused on adenotonsillectomy, reporting that in their UK hospital, almost 1000 adenotonsillar procedures were performed every year, producing 1984 kg of incinerated waste each year. Generalized to the entire UK, this would be 106,020 kg of incinerated waste per year. The authors auspicated that surgeons and operating room staff think, act and support the green changes needed.

Two studies [13,21] focused on anesthesia-related activities that contribute to operating room waste. Skowno J et al. [13] evidenced that a quarter of all medical garbage results from operating rooms, of which 25% results from anesthetic services. Therefore, great decreases in the usage of volatile anesthetics and nitrous oxide, or adoptions relating to waste management and procurement practices are auspicated. Petre MA et al. [21] administered an online survey establishing up-to-date efforts in, and barriers to, environmentally sustainable anesthesia practice to the managers of Canadian departments of anesthesia (*n* = 113). Similarly, Canadian anesthesiology residency program directors (*n* = 17) were asked to fill in an online survey defining up-to-date educational programs on environmental sustainability and recognizing interest in, and barriers to, developing a Canada-wide curriculum. Department chiefs specified that their departments contribute to sustainability efforts such as providing medical gear (65%) and recycling (58%). Despite attention to environmental sustainability, they recognized insufficient funding (72%), lack of a mandate (64%), and scarce knowledge (60%) as barriers to applying environmentally sustainable practices. Responding residency program directors asserted that residents would benefit from more education on the topic (86%) but recognized barriers involving a lack of faculty knowhow (100%), and time constraints (71%). The authors concluded that specific teaching agendas are required to deal with anesthesia-related waste management in nosocomial environments [21].

Several cost-neutral and cost-negative green approaches have been suggested, including developing periodical waste audits, digitalizing paper directives, using devices only when required, decreasing waste misclassification by instructing staff and confining medical garbage in the general waste stream, increasing recycling of paper and soft plastic packaging material, promoting green practices to encourage user consciousness and demand and promotion via professional societies to business and government [8,19,23,24]. Moreover, engagement, ownership, and training of healthcare workers in the background of a multi-disciplinary collaboration and managerial buy-in will be critical to systematizing these approaches to green the operating rooms [8]. Beloeil H et al. [14] reported that when the 5Rs rule (reduce, reuse, recycle, rethink and research) is applied, enhancement towards a decrease of the environmental footmark had been showed. Moreover, the greening of the operating theaters necessitates the commitment of all health professionals as well as other departments (pharmacy, hygiene) and management. 

Haemodialysis is another major resource-hungry procedure generating considerable quantities of waste (1.5–8 kg per dialysis); some of them represent an infectious/toxic risk for living creatures (potentially contaminated or hazardous waste), while others are damaging for the planet (plastic and non-recycled waste) [9,22]. In fact, in 2020 the Italian Society of Nephrology presented a position statement on how the environmental impact of caring for patients with kidney diseases could be decreased, suggesting, among others, reducing the burden of dialysis, encouraging the re-use of hospital devices, recycling paper, glass and non-contaminated plastic, introducing environmental-impact criteria to evaluate dialysis machines and provisions, encouraging well-organized triage of contaminated and non-contaminated devices, and demanding planet-friendly methodologies in the construction of new facilities [22].

The treatment of cataract surgery, dry eye disease and glaucoma often necessitate life-long usage of eye drops, even multiple products instilled numerous times per day. A greener approach consists of moving toward novel and safe sustained-release drug delivery systems. Indeed, the authors reported a trend over the last decade to move toward preservative-free preparations, which can be distributed by single- or multi-dose vials [7].

Diabetes Technology Society auspicated the elaboration and utilization of “green” diabetes technology [20]. The authors proposed the 5Rs rule strategy to achieve the maximum concrete benefits from one-use diabetes devices while producing the minimum quantity of waste. Indeed, the used diabetes devices intended for one-time use (injection needles, syringes, lancets, strips, blood glucose monitors, sensors, insulin bottles, infusion tubing, disposable pumps, device batteries, packaging, etc.) generate a great quantity of garbage.

Among the strategies of dealing with waste management, Sisdyani EA et al. [25] suggested that governments should prepare mandatory policies; indeed, the compulsory regulation will postulate a coercive way to induce compliant comportment. The authors interviewed 25 top managers of Indonesian public facilities, evidencing that the green behavior purpose assumes the significant usage of the four eco-control mechanisms, i.e., belief, boundary, diagnostic and interactive eco-control. Moreover, a strong intention is associated with a higher probability that the comportment will be realized through the application of boundary eco-control. 

Many studies focused on the use of mathematical algorithms in order to enhance the supply chain and medical waste management [11,12,16,18]. Indeed, medical supply chain network design is one of the critical provision difficulties that, if solved, can relieve the hazards rising from the increase of wastes [18]. Liu Z et al. [12] reported that the presentation of green governance attitudes to the study of the ideal pathway for community transitory storage, and the ideal choice of marched pathway for disposal, is efficient under certain circumstances. Focusing on several values of instrumentality, controllability, and efficiency in the operational assortment of green governance attitudes, the goal management identity, scientific algorithm use, and practical applications through simulation and experimental pathway optimization can be useful in building a green governance model [12].

The most recent studies evidenced how the COVID-19 pandemic could provisionally slow down the greening process of a nosocomial environment [7]. Indeed, the pandemic has dramatically improved the demand for N95 respirators and surgical masks across the world, leading to supply shortages, the spending of billions of dollars and production of great amounts of medical waste. Chu J et al. [10] estimated usage, costs and waste incurred by N95 respirators over the first six months of the pandemic in the USA. One N95 respirator per day per healthcare professional would require 3.29 billion respirators, cost $2.83 billion and produce 37.22 million kg of waste. A combination of a reusable respirator with H_2_O_2_ vapor-decontaminated filters would decrease costs to $831 million and produce 1.58 million kg of garbage. Decontamination and reusable respirator-based strategies could reduce the number of N95 respirators and surgical masks used, costs and garbage generated [7,10]. 

As reported by many authors, the above reported evidence and experiences are even more serious in light of the climate change that the world is facing [8,9,13,15,17,19]. Indeed, the climate crisis is documented as an influence multiplier for all the health, economic, and racial inequalities that the world already experiences in a society that has already been exposed through the COVID pandemic crisis [17].

## 4. Discussion

Our narrative review highlights how the issue of medical waste is deeply felt at an international level. In fact, the problem has been addressed by the scientific literature from different points of view and in different health contexts, trying to propose effective solutions to reduce the production of waste and improve its disposal.

Several studies focus on operating theaters, estimating how surgical activity involves 30–70% of all medical waste; in particular, endoscopy services and anesthetic procedures seem to be the most wasteful procedures [13,15,21]. Regarding endoscopy, a 2022 position paper from the Italian association of hospital gastroenterologists and digestive endoscopists (AIGO) proved how scientific societies, hospital executives and single endoscopic units can structure health policies and investments to build a “green endoscopy,” thus shaping a more sustainable health service leading to an equitable, climate-smart and healthier future [26]. The education of healthcare professionals and management appears to be a winning strategy for managing medical waste [8]. Indeed, inadequate knowledge (60%) was identified as one of the main barriers to implement environmentally sustainable practices [21]. This evidence is confirmed by other studies in literature, that agree that education of all levels in the healthcare system is important in order to drive and maintain change [27,28]. 

Hemodialysis represents a further source of medical waste that brought the Italian nephrology society in 2020 to express itself with a position paper that recommends a series of good practices in order to reduce and manage waste [22]. Indeed, a 2014 review [29] evidenced that the global efforts to combat climate change and the environmental impact of hemodialysis practice will be subjected to stricter regulations. The authors concluded that taking proactive steps, rather than being compelled by government or administration, represents a more profitable strategy. Moreover, opportunities to reduce the environmental impact of hemodialysis include capturing and reusing reverse osmosis reject water, utilizing renewable energy, and potentially reducing dialysate flow rates [30]. 

Most studies report the 5Rs rule (reduce, reuse, recycle, rethink and research) as a guide for medical waste management. Waste management can be improved through reducing the volume of waste, improving waste segregation, reusing certain medical equipment, recycling, rethinking outdated practices and dedicating time to research and the development of innovative strategies to reduce the ecological footprint on the environment [31]; moreover, novel technology, renewable energies, and smarter architectural design can be helpful in reaching the objective of waste reduction [31,32].

An aspect highlighted in the most recent studies is the impact that the COVID-19 pandemic has also had on the greening processes of healthcare facilities. Chu J et al. [10] estimated that in the first six months of the pandemic in the USA almost 40 million kg of waste were required to dismantle N95 respirators used by healthcare personnel (it must be considered that a single N95 mask contains approximately 11 g of polypropylene and/or other plastic derivatives); therefore, the process of decontamination and reusable respirator-based strategies may be able to reduce the impact of N95 respirators on the environment. Moreover, even surgical masks are mainly composed of polypropylene (PP). A 2021 review [33] estimated that the overall face mask waste generated in Peru reached 14,983,383.4 masks per day, equaling 74.9 tons of daily plastic waste, and proposed bio-based and fully degradable filters for reusable face masks in order to manage this phenomenon. A 2022 study [34] reviewed the impact of COVID-19 on the environment, reporting that during the first outbreak, Wuhan hospitals generated 240 metric tons of medical waste, compared to 50 tons previously recorded; moreover, the millions of doses of COVID-19 vaccines have led to an increase in waste, as syringes, needles, vials, personal protective equipment, and plastic and cardboard/paper-based packaging materials [34].

The request for regulation by governmental institutions is advocated by several authors, considering that mandatory regulation should provide a coercive way to force compliant behavior [25]; indeed, scientific literature reported many common barriers concerning the implementation of greening strategies, as well as a lack of leadership, perceived risk of infection, lack of data, concerns about increased workload, staff attitudes, and resistance to change [31]. Therefore, associating clear and restrictive rules with education campaigns for health personnel could accelerate greening processes.

Many waste management models are described in the literature. Blessy J et al. [35] focused on non-infected plastic wastes generated at healthcare facilities, proving that this type of waste is either disposed of in landfills or inadequately incinerated. Recycling of plastics is the proposed solution to deal with such medical waste and integrating current recycling strategies with new sustainable alternatives. Moreover, a better awareness about recycling possibilities among healthcare workers and the commitment to collect and recycle plastic waste is required, and plastic devices should be designed to be more easily recyclable. The management of contaminated plastic waste during the COVID-19 pandemic was investigated in a 2022 study [36]. The authors reported many methods that can be used to sterilize waste before recycling treatments, such as focusing on microbial degradation and the use of microorganisms for the digestion of plastic polymers, as this novel method assures ecological advantages, cost-effectiveness, ease of use and maintenance. Another innovative approach to deal with the increase of polypropylene (PP)-based PPE for healthcare personnel that was proposed is the recovery of these plastics for the production of fuel-like liquid oil and solid char through thermal decomposition via pyrolysis process. This method reduces PP plastic waste and produces pyrolysis liquid oil and solid char to be used in fuel applications [37].

The results of our study must be read from an environmental point of view, but also from an economic point of view. Indeed, hospitals generate on average 1.5 billion kg of solid waste annually, relying on the $US 40.3 billion disposable medical supply industry. Many studies focused on the cost-effectiveness of waste management initiatives in nosocomial environments and estimated that reducing, recycling and reusing tools and instruments leads to savings of thousands of dollars a year, even if an initial investment is required to optimize processes [31]. A recommendation to raise awareness among managers on this issue could be the addition of the purchase price of an item to the cost of its waste disposal, occupational health costs, environmental impact, and warehousing costs to determine the ultimate cost of purchasing the disposable medical item [38].

The limitations of this study include the analysis of limited previous literature and poor data sources, given that most references consisted of opinion papers, letters and small case studies. Nevertheless, this study stands out as the most comprehensive overview of the topic up-to-date and encompasses various sustainability topics. Moreover, it focuses on the impact of the COVID-19 pandemic on waste management and green hospital topics. Future studies are needed to expand ongoing and new sustainability projects, testing cost-effectiveness and associated impact on patient safety and healthcare.

## 5. Conclusions

Given the multifactorial nature of this issue and the measures to be implemented to manage it, a multifactorial and multistep approach is needed. In the short term, health facilities should become aware of the waste produced, in quantitative and qualitative terms, and of the resources used in order to define the main critical issues, and define the methods of disposal and possible recycling. In the middle term, the creation of an “environmental greening team” to increase knowledge, improve attitudes and facilitate the success of green initiatives is suggested [32]. From a public health point of view, the implementation of these teams within the health directorates could be useful for the management of all environmental aspects of a health facility (waste, water, air quality, etc.), as well as the implementation of cost-effective measures that would guarantee savings (even in the medium-long term) on the budgets of health nosocomial environments. In the long term, the environmental weight of waste and medical procedures must be considered in the company’s finances (even when purchasing material) and their correct management must be included in the objectives of the healthcare facilities’ top managers.

Finally, the aspects described above must be read in the actual historical contest in which climate change is manifesting itself as one of the main problems that humanity is now facing [39]. Therefore, the above-described multifactorial approach is necessary in order to manage medical waste and make it economically advantageous to convert into strategies aimed at reducing it. With respect to governmental institutions, clear and restrictive rules must be promulgated and applied, which encourage (also economically) the greening of hospitals. Organizations such as Practice Greenhealth and Health Care Without Harm can be useful as educational and collaborative resources for the exchange of ideas [32].

## Figures and Tables

**Figure 1 ijerph-19-15812-f001:**
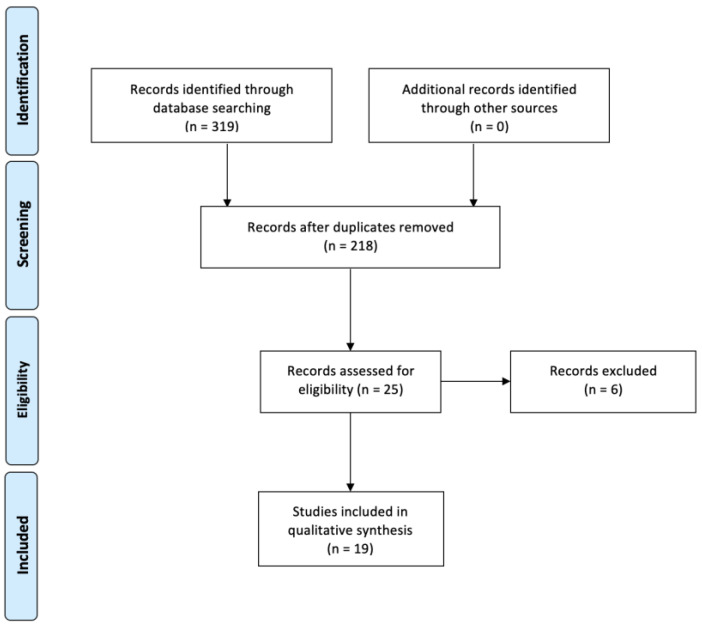
Flow-chart of the bibliographic research.

**Table 1 ijerph-19-15812-t001:** Included studies.

Author	Year	Study Model	Country	Ref.
Giannaccare, G.	2022	Letter	Italy	[7]
Vacharathit, V.	2022	Observational study	USA	[8]
Burnier, M.	2021	Editorial	France	[9]
Chu, J.	2021	Observational study	USA	[10]
Deepak, A.	2021	Observational study	India	[11]
Liu, Z.Y.	2021	Observational study	China	[12]
Skowno, J.	2021	Article	Australia	[13]
Beloeil, H.	2021	Observational study	France	[14]
De Melo, S.V.	2021	Observational study	USA	[15]
Khahro, S.H.	2021	Article	Saudi Arabia	[16]
Reynolds, P.	2021	Letter	USA	[17]
Alizadeh, M.	2020	Article	Iran	[18]
Gill, A.S.	2020	Letter	UK	[19]
Klonoff, D.C.	2020	Editorial	USA, Germany	[20]
Petre, M.A.	2020	Observational study	Canada	[21]
Piccoli, G.B.	2020	Position paper	Italy	[22]
Raymond, S.B.	2020	Commentary	USA	[23]
Shum, P.L.	2020	Observational study	Australia	[24]
Sisdyani, E.A.	2020	Observational study	Indonesia	[25]

## Data Availability

Not applicable.

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
