# Peer review of "Waste Management and the Perspective of a Green Hospital—A Systematic Narrative Review"

_ijerph, 2022, doi:10.3390/ijerph192315812_

Round 1
Reviewer 1 Report
The study's main objective is to systematically review the experiences acquired worldwide in nosocomial settings related to managing healthcare waste. Nineteen studies, selected between January 2020 and April 2022 on Scopus, MEDLINE/PubMed and Web of Science databases, were included in the systematic narrative review. Operating room and hemodialysis activities seem to be the procedures more associated to waste production. The 5Rs rule (reduce, reuse, recycle, rethink and research) was a strategy to derive the maximum practical benefit from disposable diabetes devices while generating the minimum amount of waste. The COVID-19 pandemic slowed down the greening process of the nosocomial environment. The climate change that the world is experiencing requires a multifactorial approach to dealing with medical waste management. Education of health personnel and managers, regulation by governmental institutions, creation of an “environmental greening team”, and awareness of stakeholders and policymakers are some of the measures needed to aim for greening healthcare facilities. The study is impressive and has achieved its stated objectives of the study. I have minor comments on the paper, i.e.,
1) Include a few current literature reviews related to the theme of the study. 2) Include possible research questions in the introduction section.
3) To add the study's research objectives in line with the research questions. 4) To identify possible research gap(s) and how the study filled these gap(s) by their study, and
5) To add a few short-term, medium-term and long-term policy implications for establishing green hospitals.
Author Response
Q1. Include a few current literature reviews related to the theme of the study.
A1. We have revised the manuscript according to your suggestions.
Q2. Include possible research questions in the introduction section, to add the study's research
objectives in line with the research questions, to identify possible research gap(s) and how the study
filled these gap(s) by their study.
A2. We revised the objectives paragraph.
Q3. To add a few short-term, medium-term and long-term policy implications for establishing green
hospitals.
A3. We revised the conclusions paragraph.
Reviewer 2 Report
this paper has significance for green health care as of its head of green health-care or green hospital.
but it seems only making some suggestions through literatures investigations.
the abstract content is wide or not concentrated,the conclusion is too long without enough supports
some managing waster models, collecting water routes, treating them, etc. are required to add.
some typical waster managing examples such as polymer materials or composite materials are required.
in covid 19, n95 masks are only one of the masks for populations. their statistical data may show the process of calculations.
Author Response
Q1. this paper has significance for green health care as of its head of green health-care or green
hospital. but it seems only making some suggestions through literatures investigations.
A1. We have revised the manuscript according to your suggestions.
Q2. the abstract content is wide or not concentrated, the conclusion is too long without enough
supports
A2. Revised.
Q3. some managing waster models, collecting water routes, treating them, etc. are required to add.
A3. We added few examples on discussion.
Q4. some typical waster managing examples such as polymer materials or composite materials
are required.
A4. We added few examples on discussion.
Q5. In covid 19, n95 masks are only one of the masks for populations. Their statistical data may
show the process of calculations.
A5. Revised.